# Structures of LIG1 that engage with mutagenic mismatches inserted by polβ in base excision repair

Qun Tang[1], Mitchell Gulkis [1], Robert McKenna[1] & Melike Çağlayan [1✉]

DNA ligase I (LIG1) catalyzes the ligation of the nick repair intermediate after gap filling by DNA polymerase (pol) β during downstream steps of the base excision repair (BER) pathway. However, how LIG1 discriminates against the mutagenic 3′-mismatches incorporated by polβ at atomic resolution remains undefined. Here, we determine the X-ray structures of LIG1/nick DNA complexes with G:T and A:C mismatches and uncover the ligase strategies that favor or deter the ligation of base substitution errors. Our structures reveal that the LIG1 active site can accommodate a G:T mismatch in the wobble conformation, where an adenylate (AMP) is transferred to the 5′-phosphate of a nick (DNA-AMP), while it stays in the LIG1-AMP intermediate during the initial step of the ligation reaction in the presence of an A:C mismatch at the 3′-strand. Moreover, we show mutagenic ligation and aberrant nick sealing of dG:T and dA:C mismatches, respectively. Finally, we demonstrate that AP-endonuclease 1 (APE1), as a compensatory proofreading enzyme, removes the mismatched bases and interacts with LIG1 at the final BER steps. Our overall findings provide the features of accurate versus mutagenic outcomes coordinated by a multiprotein complex including polβ, LIG1, and APE1 to maintain efficient repair.

---

[1] Department of Biochemistry and Molecular Biology, University of Florida, Gainesville, FL 32610, USA. ✉email: caglayanm@ufl.edu

Human DNA ligases catalyze phosphodiester bond formation between 5′-phosphate (P) and 3′-hydroxyl (OH) termini on the ends of broken DNA strands using a high-energy cofactor ATP and $Mg^{2+}$ in three chemical and sequential steps: (i) nucleophilic attack on ATP by the ligase and formation of a covalent intermediate in which an adenylate (AMP) is linked to an active site lysine (LIG-AMP), (ii) transfer of the AMP to the 5′-phosphate-terminated DNA strand to form a DNA-AMP intermediate, and (iii) ligase-catalyzed attack by 3′-OH of the nick on DNA-AMP to join adjacent 3′-OH and 5′-P ends and liberate AMP[1–5]. The accuracy of the nick sealing reaction relies on the formation of a Watson–Crick base pair between the 3′-OH and 5′-P ends that requires high fidelity DNA synthesis by DNA polymerase[6]. Mismatched nucleotides inserted by DNA polymerases can cause base substitutions, and in the case of no proofreading, this could lead to genomic instability and human diseases[7–9]. In the presence of 3′-damaged or modified ends, DNA ligases can fail, resulting in the formation of 5′-adenylated-DNA (5′-AMP), also referred to as an abortive ligation product[10]. However, despite the importance of nick sealing at the end of almost all DNA repair pathways as well as DNA replication, the mechanism of mismatch discrimination by which a human DNA ligase can encounter mutagenic DNA ends during the three steps of the ligation reaction remains unknown. Because nicks are potentially deleterious DNA lesions that may lead to the formation of lethal double-strand breaks, a thorough understanding of the nick sealing mechanism is crucial for a comprehensive understanding of genome maintenance[3,5].

Base excision repair (BER) requires tight coordination that entails the substrate-product channeling of DNA intermediates between repair proteins so that the release and accumulation of toxic and mutagenic single-strand break intermediates are minimized in cells[11–13]. The downstream steps of the BER pathway involve DNA synthesis by DNA polymerase (pol) β and subsequent nick sealing by DNA ligase I (LIG1) or IIIα to complete the repair of small single-base DNA lesions[10]. Polβ, an error-prone polymerase without 3′−5′ proofreading activity, can incorporate mismatch nucleotides at a frequency of 1 in ~5000 during template-directed DNA synthesis[14]. Furthermore, several of the cancer-associated polβ variants possess aberrant repair functions, such as reduced fidelity stemming from impaired discrimination against incorrect nucleotide incorporation, and expression of these variants in cells induces cellular transformation and genomic instability[15,16]. In our studies, we have found that polβ mismatch or oxidized nucleotide 7,8-dihydro-8′-oxo-dGTP (8-oxodGTP) insertion products can generate a problematic nick repair intermediate for the subsequent ligation step in the BER pathway[6,10,17–28]. These findings contribute to understanding the important molecular determinants that ensure accurate BER pathway coordination or result in the impaired handoff from polβ to DNA ligase at the downstream steps. Nevertheless, the extent to which discrimination by a DNA ligase counteracts mutagenic repair products during the final nick sealing step of the BER pathway remains unknown. Particularly, it is unclear how LIG1 dictates accurate versus mutagenic outcomes while engaging with polβ-mediated base substitution errors at atomic resolution.

In the present study, we defined the molecular basis of the human LIG1 mismatch discrimination mechanism via moderate resolution structures of LIG1/nick DNA duplexes containing A:C and G:T mismatches at the 3′-strand. Our structures revealed that the LIG1 active site can accommodate G:T mismatch in the wobble conformation, where the ligase validates mutagenic G:T ligation during the adenyl transfer step of the ligation reaction (DNA-AMP). We also captured LIG1 in complex with nick DNA harboring Watson–Crick A:T base pair at this second step.

However, we found that the ligase active site lysine (K568) stays adenylated (LIG1-AMP) while engaging with a A:C mismatch, which refers to the first ligation step. Furthermore, we showed mutagenic and defective nick sealing of 3′-G:T and 3′-A:C mismatches, respectively. Finally, our results demonstrate that AP-endonuclease 1 (APE1), a complementary DNA-end processing enzyme, removes a mismatched base (3′-dG or 3′-dA) from the nick DNA substrates via its exonuclease activity and coordinates with LIG1 for mismatch removal coupled to DNA ligation at the final steps of the BER pathway. Our overall results demonstrate the strategies by which LIG1 engages with the base substitution errors incorporated by polβ and reveal the requirement of a multiprotein assembly (polβ, LIG1, and APE1) to maintain repair efficiency.

## Results

**Structures of LIG1/nick repair intermediates with mismatched DNA ends.** To elucidate the ligase strategies that deter or favor the ligation of repair intermediates that mimic polβ mismatch insertion products at atomic resolution, we determined the structures of LIG1/nick DNA duplexes containing G:T and A:C mismatches as well as cognate A:T base pair at the 3′-strand (Table 1 and Fig. 1).

In the structure of LIG1 bound to a nick DNA duplex containing a cognate A:T base pair, the 5′-terminus was adenylated, and a DNA-adenylate (DNA-AMP) intermediate was observed (Fig. 1a). Similarly, we showed that the ligase active site could accommodate a G:T mismatch during step 2 of the ligation reaction when AMP is transferred to the 5′-P of nick DNA (Fig. 1b). The superimposition of LIG1 structures with G:T

**Table 1 X-ray data collection and refinement statistics of LIG1/nick DNA duplexes containing cognate A:T, and mismatched G:T and A:C ends at the 3′-strand.**

| PDB entry ID | LIG1[EE/AA] A:T 7SUM | LIG1[EE/AA] G:T 7SXE | LIGI[EE/AA] A:C 7SX5 |
|---|---|---|---|
| *Data collection* | | | |
| Space group | $P2_12_12_1$ | $P2_12_12_1$ | $P2_12_12_1$ |
| *Cell dimensions* | | | |
| a, b, c (Å) | 65.255 116.59 124.16 | 64.254 116.00 125.63 | 63.755 116.41 126.01 |
| a, b, g (°) | 90 | 90 | 90 |
| Resolution (Å) | 20–2.90 (2.95–2.90) | 20–3.0 (3.05–3.0) | 20–2.8 (2.85–2.8) |
| $R_{sym}$ | 0.126 (0.868) | 0.173 (1.425) | 0.188 (0.759) |
| $I / \sigma (I)$ | 21.2 (2.8) | 17.7 (1.5) | 12.5 (1.7) |
| $CC_{1/2}$ | 0.995 (0.827) | 0.988 (0.733) | 0.947 (0.707) |
| $CC^*$ | 0.999 (0.951) | 0.997 (0.920) | 0.986 (0.910) |
| Completeness (%) | 99.9 (99.7) | 99.8 (99.5) | 99.0 (98.1) |
| Redundancy | 12.2 (11.9) | 11.4 (11.2) | 6.3 (6.1) |
| *Refinement* | | | |
| Resolution (Å) | 20–2.90 | 20–3.0 | 20–2.8 |
| No. reflections | 21183 | 19238 | 22098 |
| $R_{work}/R_{free}$ | 0.182/0.227 | 0.204/0.245 | 0.175/0.212 |
| *Non-H atoms* | | | |
| Protein/DNA | 5752 | 5491 | 5728 |
| AMP | 22 | 22 | 23 |
| Solvent | 141 | 38 | 182 |
| *Average B-factors ($Å^2$)* | | | |
| Protein/DNA | 54.28 | 85.58 | 38.91 |
| AMP | 40.62 | 87.39 | 32.12 |
| Solvent | 42.28 | 44.52 | 29.70 |
| *R.M.S.D* | | | |
| Bond lengths (Å) | 0.013 | 0.013 | 0.012 |
| Bond angles (°) | 1.41 | 1.42 | 1.36 |

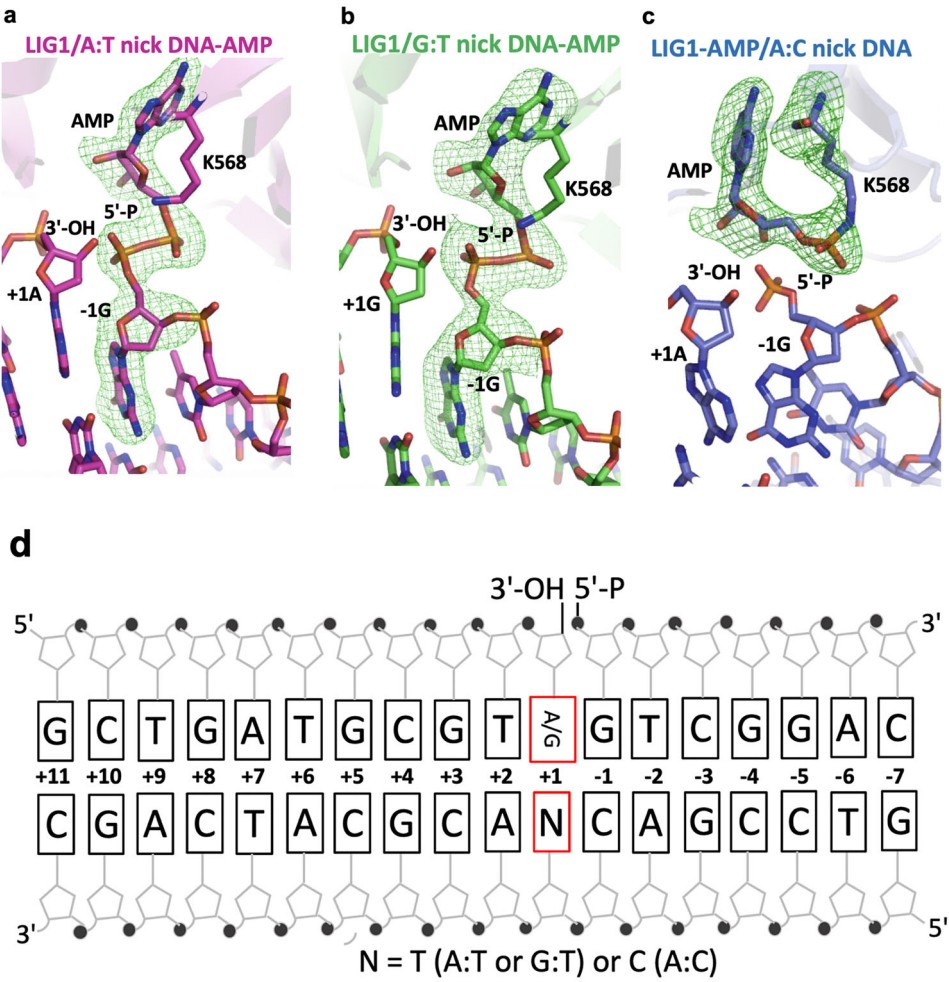

**Fig. 1 Structures of LIG1 bound to nick DNA duplexes containing G:T and A:C mismatches. a–c** X-ray crystal structures of LIG1/nick DNA duplexes containing cognate A:T (**a**) and mismatched G:T (**b**) and A:C (**c**) ends at the 3′-strand. Simulated annealing omit maps (Fo-Fc) of LIG1/A:T (magenta) and G:T (green) structures with the AMP-DNA complex are displayed for the adenylated 5′-P of the nick (DNA-AMP) contoured at 3σ. Simulated annealing omit maps (Fo-Fc) of LIG1/A:C (blue) with the ligase-AMP complex are displayed for adenylated LIG1 (LIG1-AMP) at the K568 active site residue contoured at 3σ. DNA and LIG1 are shown as sticks and cartoons, respectively, and the map is depicted in green. **d** Schematic view of DNA used in the LIG1/nick DNA duplex crystallization.

mismatch and A:T base pair revealed no significant differences, with a superimposed Cα root mean square deviation of 0.609 Å. In both structures, the LIG1/nick conformation was poised for nick sealing during step 3 of the ligation reaction. However, in the structure of the LIG1/nick DNA duplex containing an A:C mismatch, we showed that the ligase active site exhibited the LIG1-adenylate conformation, where the active side lysine residue (K568) was covalently bound to the AMP phosphate (Fig. 1c). This refers to step 1 of the ligation reaction. Our results indicate that LIG1 stays in its initial adenylated state and cannot move forward with subsequent adenyl group transfer to the 5′-P on the downstream strand to activate the ligase for attack by the upstream 3′-OH of the nick DNA (Fig. 1d). These observations suggest that the A:C base pair imparts the active site conformation that suppresses the further chemical steps of catalysis. Overall, our LIG1/mismatch structures demonstrate that the ligase is trapped as the adenylated-DNA intermediate (AMP-DNA), which favors the mutagenic ligation of the G:T mismatch, and the active site remains in an inactive conformation (AMP-LIG1), which deters nick sealing of the A:C mismatch.

We observed a Watson–Crick conformation with two hydrogen bonds for cognate A:T base pairs, while G:T and A:C mismatches exhibited the wobble conformation (Fig. 2a). The

superimposition of the LIG1 A:T, G:T, and A:C structures showed differences in the positions of the mismatched termini at the 3′-OH end of nick DNA (Fig. 2b, c). Furthermore, our LIG1/mismatch structures revealed that the ribose adopted a different sugar pucker depending on the identity of the 3′-OH base pair, cognate versus mismatched. We observed the C3′-endo pucker in the LIG1/A:T structure while the LIG1/G:T and LIG1/A:C mismatch structures exhibited the C4′-exo sugar pucker (Supplementary Fig. 1). In addition, similar to the previously reported LIG1 structures[29–32], LIG1 containing the DNA binding (DBD), adenylation (AdD), and oligonucleotide binding (OBD) domains completely encircles the nick DNA containing the cognate or mismatched ends (Supplementary Figs. 2 and 3).

**LIG1 active site shows distinct DNA conformations depending on the identity of the mismatch.** Our LIG1/mismatch structures demonstrate that the ligase active site exhibits distinct mismatch-specific conformations and shows significant differences in the positions of the 5′-P and 3′-OH strands at the nick around the upstream and downstream DNA (Fig. 3a). In the G:T and A:T structures showing the formation of the 5′-5′ phosphoanhydride AMP-DNA intermediate, we observed that 5′-P is closer to the

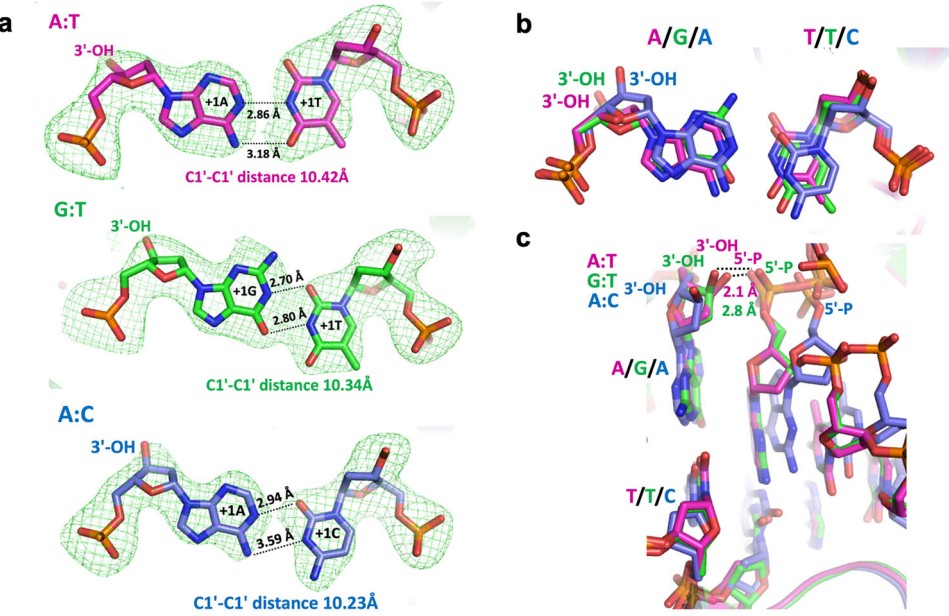

**Fig. 2 Base-pairing architecture of mismatched ends at the 3′-OH terminus. a** Simulated annealing omits maps (Fo-Fc, 3σ) that are depicted in green show the C1′-C1′ distances of A:T (magenta), G:T (green) and A:C (blue) base pairs. **b, c** Overlay of LIG1 structures shows differences in the positions of the 3′-OH terminus (**b**) and distances between DNA ends of a nick (**c**).

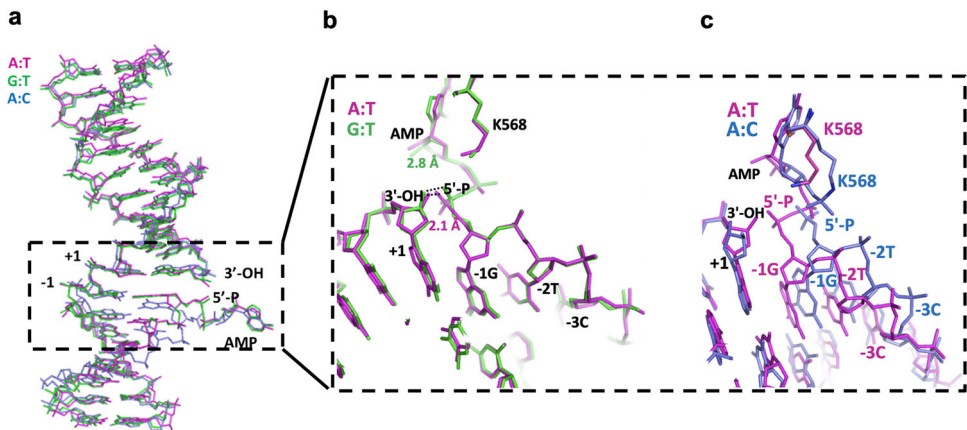

**Fig. 3 LIG1/mismatch structures exhibit different DNA conformations. a** Superimposition of A:T (magenta), G:T (green) and A:C (blue) structures shows the positions of the 5′-P and 3′-OH ends of the nick. **b, c** Overlay of A:T/G:T (magenta/green) and A:T/A:C (magenta/blue) structures shows the positions of DNA-AMP and AMP-K568.

3′-OH strand of the nick to ensure proper positioning and to seal the phosphodiester backbone (Fig. 3b). The distances between the ends of the nick DNA for A:T and G:T were 2.1 Å and 2.8 Å, respectively. In the LIG1/A:C structure, the 3′-OH of the nick DNA was rotated 50° from that of LIG1/A:T. The overlay of both structures demonstrated significant differences in the conformations of the 5′-strand due to clear shifts in the positions of the −1G, −2T, and −3C nucleotides relative to the upstream DNA (Fig. 3c).

Moreover, the position of phenylalanine (Phe) at 872 (F872), which is located upstream of the nick and positioned close to the deoxyribose moiety of the nucleotide at the 5′-end, shows an important difference between the LIG1/A:T and A:C structures (Fig. 4). The overlay of both LIG1 structures demonstrated that the F872 distorts the alignment upstream of the A:T nick, where −1G and +1 A nucleotides are in parallel between the 3′ and 5′ strands (Fig. 4a). Notably, superimposition of the structures in placement of the wobble A:C mismatch and the cognate A:T base

pair at the active site of LIG1 demonstrated the shift in the position of F872 (Fig. 4b). As reported in the previously solved LIG1 structures[29–32], F635 and F872, which are located on the AdD and OBD domains of LIG1, respectively, are forced into the minor groove and play important roles in the catalysis[33]. However, in the LIG1/A:T and G:T structures, we did not observe a difference in the position of F635, which was located near the upstream end, where it was positioned near the sugar moiety of the 3′-end (Supplementary Fig. 4). The interaction interface between R874 and the −2T nucleotide of the downstream DNA showed a change in the LIG1/A:T versus A:C structures (Fig. 5a). In both LIG1/mismatch structures, we found conformational differences at the active site residues Arg(R)589 and Leu(L)544, which were positioned close to the 5′-P end of the nick (Fig. 5b). Similarly, the distance between the R589 and L544 side chains was shifted because of the differences in the position of AMP, forming DNA-AMP and K568-AMP, as shown in the overlay of the LIG1/A:T and A:C structures, respectively (Fig. 5b).

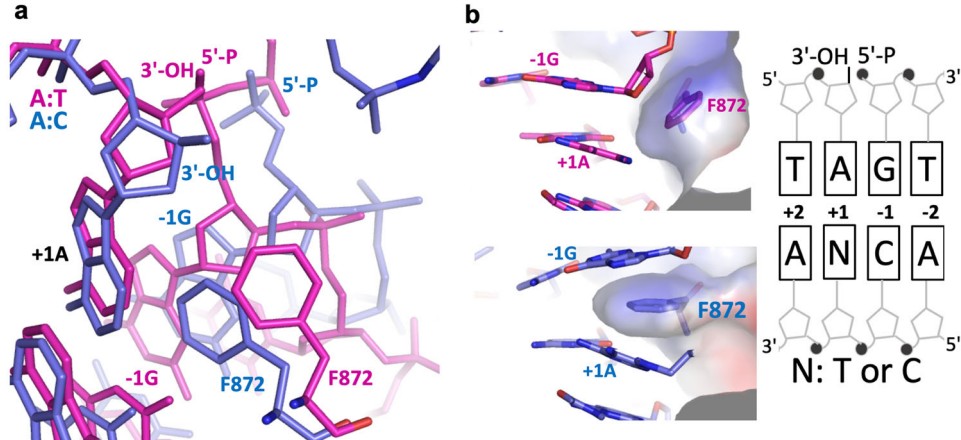

**Fig. 4 LIG1 active site demonstrates a shift in the position of the F872 residue in the placement of the A:C mismatch. a** Overlay of LIG1 structures bound to the nick DNA duplexes containing cognate A:T (magenta) and mismatched A:C (blue) ends shows the conformational change at the F872 as sticks. **b** Surface representations of the F872 residue show the different positions at the +1A and −1G bases of nick DNA.

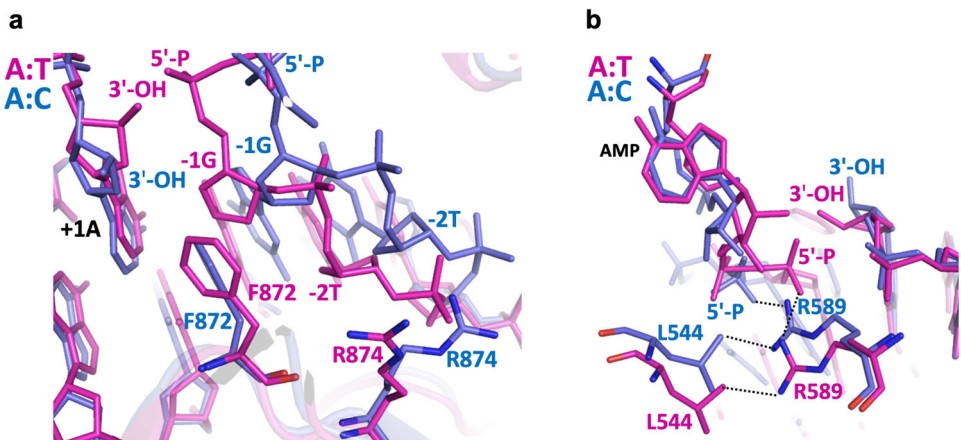

**Fig. 5 Positions of LIG1 active site residues exhibit differences depending on the identity of the mismatch at the 3′-OH terminus. a, b** Overlay of LIG1 structures bound to the nick DNA duplexes containing A:T (magenta) and A:C (blue) ends shows the positions of F872 and R874 (**a**) and R589 and L544 (**b**). The residues F872 and the neighboring R874 make direct DNA contacts with nucleotides −1G and −2T, respectively. R589 is positioned close to the 5′-P end and makes contacts with the L544 side chain.

**Efficiency of downstream BER steps involving mismatch-containing repair intermediates.** To understand the efficiency of the downstream steps of the BER pathway and to investigate the substrate discrimination of LIG1 against nick DNA substrates in the presence of G:T and A:C mismatches, we performed BER assays in vitro (Supplementary Fig. 5).

We first evaluated the efficiency of the LIG1 wild-type or EE/AA mutant using the nick DNA substrates containing 3′-preinserted dG:T and dA:C mismatches (Fig. 6 and Supplementary Fig. 6). For both proteins, we observed efficient ligation of the 3′-dG:T mismatch (Fig. 6a, b; lanes 9−14). This was very similar to the ligation products of 3′-dA:T nick DNA in the control reactions (Fig. 6a, b; lanes 2−7). However, the end-joining efficiency of LIG1 was diminished in the presence of dA:C mismatch at the 3′-end (Fig. 6a, b; lanes 16–21). There was ~30- to 90-fold difference in the amount of ligation products between dG:T and dA:C mismatches with wild-type (Fig. 6c) and EE/AA (Fig. 6d) mutant of LIG1 (Supplementary Fig. 7). For both proteins, we observed the DNA intermediates with 5′-adenylate (AMP) in the presence of the 3′-mismatches (Supplementary Fig. 8).

We then evaluated the efficiency of polβ gap filling and subsequent ligation steps in the coupled assays. For this purpose,

we tested polβ dGTP:T and dATP:C mismatch nucleotide insertion coupled to DNA ligation at the same time points in reactions including both polβ and LIG1 (wild-type or EE/AA mutant). In these assays, we used one nucleotide gap DNA substrates with a template C or T (Fig. 7 and Supplementary Fig. 9). Our results showed that the repair products after polβ dGTP:T insertion were ligated efficiently by wild-type LIG1 (Fig. 7a; lanes 7−10). These products were similar to the ligation products of the nick repair intermediate after polβ dGTP:C insertion in the control reactions (Fig. 7a; lanes 2–5). The amounts of ligation products were relatively lower for polβ dGTP:T mismatch insertion than for dGTP:C correct nucleotide insertion (Fig. 7b). However, we did not observe ligation products in the reaction of polβ dATP mismatch insertion opposite C (Fig. 7a; lanes 12−15). This could have been due to inefficient A:C mismatch insertion and reduced base substitution fidelity of polβ, as reported for all possible incorrect base pairings[34–36]. For the LIG1 EE/AA mutant, we obtained similar results showing the efficient ligation of polβ dGTP:C and dGTP:T insertion products (Fig. 7c; lanes 2−5 and 7-10, respectively). However, in the presence of the low-fidelity LIG1, the products of self-ligation (i.e., end joining of one nucleotide gap DNA itself) appeared simultaneously with the complete ligation of nicked polβ

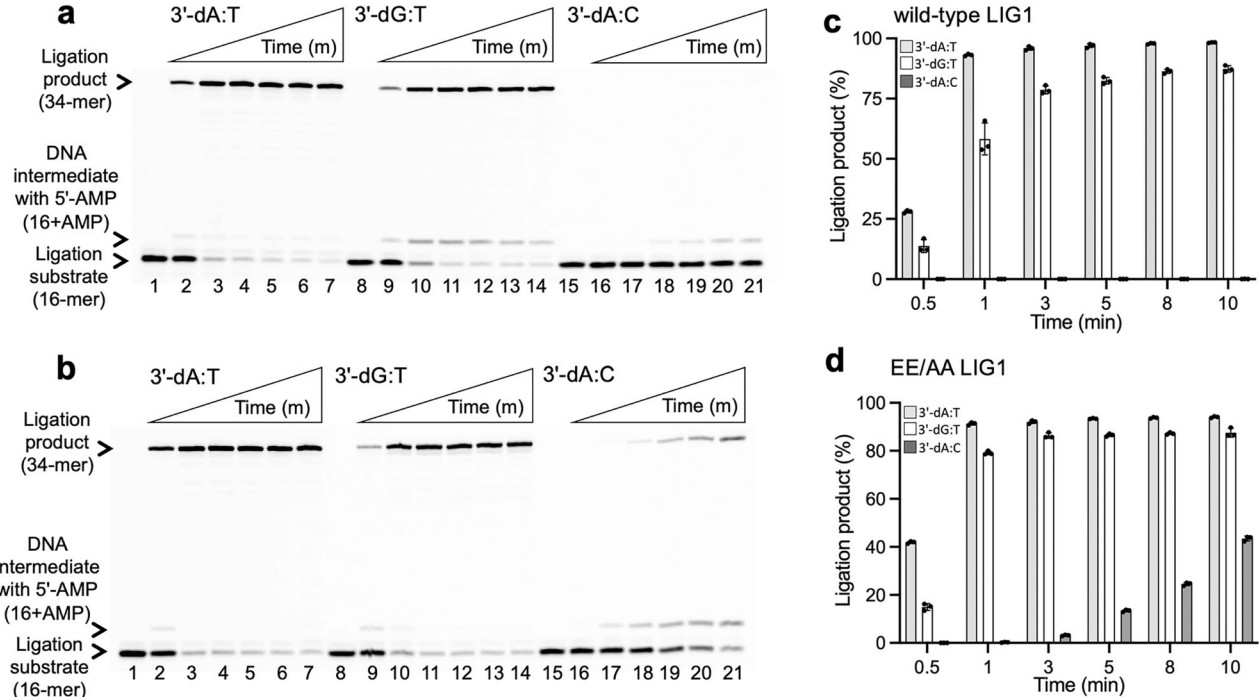

**Fig. 6 Ligation of nick repair intermediate with G:T and A:C mismatches by LIG1. a, b** Lanes 1, 8, and 15 are the negative enzyme controls of the nick DNA substrates containing 3′-preinserted dA:T, dG:T, and dA:C, respectively. Lanes 2–7, 9–14, and 16–21 are the reaction products for nick sealing of DNA substrates with 3′-preinserted dA:T, dG:T, and dA:C, respectively, by wild-type (**a**) and EE/AA mutant (**b**) LIG1 and correspond to time points of 0.5, 1, 3, 5, 8, and 10 min. **c, d** The graphs show time-dependent changes in the amounts of ligation products. The data represent the average from three independent experiments ±SD.

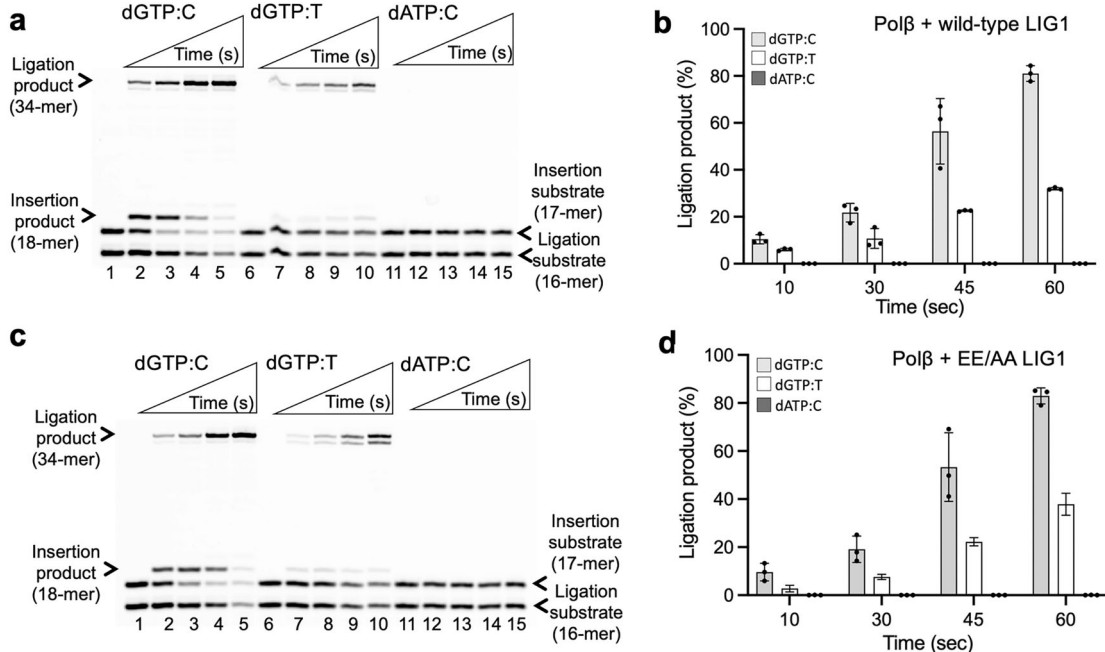

**Fig. 7 Ligation of polβ mismatch nucleotide insertion products by LIG1. a, b** Lanes 1, 6, and 11 are the negative enzyme controls of the one nucleotide gap DNA substrate with templates C, T, and C, respectively. Lanes 2–5, 7–10, and 12–15 are the reaction products for the ligation of polβ dGTP:C, dGTP:T, and dATP:C insertions by wild-type (**a**) and EE/AA mutant (**b**) LIG1, respectively, and correspond to time points of 10, 30, 45, and 60 sec. **c, d** The graphs show time-dependent changes in the amounts of ligation products. The data represent the average from three independent experiments ± SD.

insertion products (Fig. 7c, compare lines 5 and 10), as we have reported previously[24,27]. The amounts of ligation products after both polβ insertions were also found to be similar to those with the wild-type enzyme (Fig. 7d and Supplementary Fig. 10).

**Interplay between APE1 and LIG1 during processing of the nick repair intermediates with mismatched ends.** DNA repair intermediates with a damaged or mismatched base at the 3′-end can block pathway coordination and become persistent DNA strand breaks if not repaired[37]. It has been reported that APE1, the BER protein involved in the initial steps of the repair pathway, can act as a proofreader of polβ errors and remove a mismatched or damaged base from the nick repair intermediates[38]. To investigate the processing of mutagenic nick repair products with mismatched bases at the 3′-end, we examined the role of APE1 as a compensatory DNA end-processing enzyme. For this purpose, we first evaluated the 3′-5′ exonuclease activity of APE1 in the reaction mixture that included nick DNA substrates containing 3′-preinserted dG:T and dA:C mismatches. Furthermore, we investigated the processing of the nick DNA substrates with 3′-dG:T and 3′-dA:C mismatches in the coupled reactions, including both APE1 and LIG1, to test the efficiency of mismatch removal and ligation simultaneously (Supplementary Fig. 11).

We did not observe a significant difference in the mismatch base removal efficiency of APE1 between 3′-dG:T and 3′-dA:C mismatches (Fig. 8a). Our results demonstrated that APE1 could remove 3′-dG and 3′-dA bases from nick DNA substrates with template bases of T and C, respectively (Supplementary Fig. 12). In the coupled assays, APE1 mismatch removal products accumulated along with the ligation products for the nick DNA substrates with 3′-dG:T mismatch (Fig. 8b; compare lanes 2 and 3–7). However, we mainly observed the products of 3′-dA mismatch removal by APE1 from the nick DNA substrate with 3′-dA:C (Fig. 8b; compare lanes 9 and 11–14, Supplementary Fig. 13).

Finally, we quantitatively monitored the real-time kinetics of the protein-protein interaction between APE1 and LIG1 by surface plasmon resonance (SPR) assay. In this assay, the interacting protein partner of APE1 was immobilized on CM5 biosensors onto which the LIG1 protein was passed as an analyte. Our results showed protein-protein interaction with an equilibrium binding constant ($K_D$) of 117 nM between APE1 and LIG1 (Fig. 8c). In previously published studies, physical interactions for APE1 have been reported for BER proteins such as DNA glycosylase, polβ, and XRCC1[33,39]. Thermodynamic and domain mapping studies have also shown that polβ interacts with the N-terminal noncatalytic region of LIG1[40]. Overall, our results indicate that the multiprotein repair complex containing polβ, APE1, and LIG1 can determine the efficiency of BER at the downstream steps when a nick repair intermediate with an incompatible end is generated due to polymerase-mediated mutagenic mismatch insertions.

## Discussion

Spontaneous mutagenesis from DNA replication errors has been a prominent source of base substitution errors in the tumor suppressor genes in different types of cancer[41]. The intrinsic formation of Watson–Crick like mismatches has been reported as an important determinant of the base substitution mutagenesis such as mismatches G:T and A:C that sit well within the dimensions of the DNA double helix to maintain the geometry of a Watson–Crick base pair[42]. The effects of the A:C mismatch on helix stability and dynamics differ from those of the G:T base pair[43]. The G:T is the most common mismatch arising from the deamination of 5-methylcytosine to thymine and the conformation and dynamics of G:T mismatch (wobble versus Watson–Crick) are the best studied base pair by X-ray crystallography and NMR relaxation dispersion studies in DNA and RNA duplexes[44–46]. Our study represents the LIG1 structures for these particular G:T and A:C mismatches at the ligase active site.

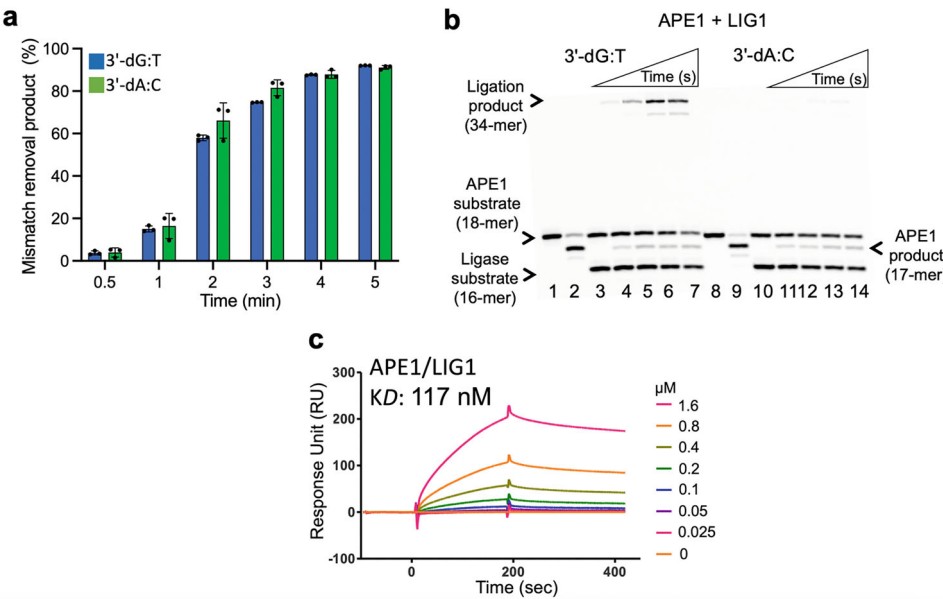

**Fig. 8 APE1 and LIG1 interact on the repair intermediate containing G:T and A:C mismatches. a** APE1 activity on the removal of the mismatched bases from the nick repair intermediates with 3′-preinserted dG:T and dA:C mismatches. The graph shows time-dependent changes in the amounts of APE1 excision products. The data represent the average from three independent experiments ± SD. **b** APE1 mismatch removal coupled to DNA ligation by LIG1. Lanes 1 and 8 are the negative enzyme controls, and lanes 2 and 9 are APE1 mismatch removal products for the nick DNA substrates with 3′-preinserted dG:T and dA:C, respectively. Lanes 3 and 10 are the negative enzyme controls. Lanes 4–7 and 11–14 are the reaction products for APE1 mismatch removal and nick sealing of DNA substrates with 3′-preinserted dG:T and dA:C, respectively, and correspond to time points of 10, 30, 45, and 60 sec. **c** Real-time protein-protein interaction analysis between APE1 and LIG1 by SPR assay.

DNA polymerases employ a series of pre-chemistry conformational checkpoints to exclude the incorporation of mismatches[47]. The mechanisms of mismatch discrimination by DNA polymerases from diverse families and organisms have been extensively reported through structural studies with a series of mismatches at the polymerase active site (Supplementary Table 1). In particular, G:T, T:G, A:C, and C:A mismatches are able to adopt a multitude of distinct conformations with exquisite sensitivity to factors such as their location, electrostatic environment of the polymerase active site, the identity of the catalytic metal ion ($Mg^{2+}$ or $Mn^{2+}$), correct downstream dNTP binding and strand specificity[48–61]. They have been hypothesized to avoid discrimination by DNA polymerases due to a Watson–Crick-like (WC-like) base pairing conformations caused by ionization or tautomerization of the nucleobases, which leads to a base pair with the proper geometry and base stacking for catalysis[48–57]. The first structural evidence of a WC-like conformation was shown in the crystal structure of human X family polλ between an incoming nonhydrolyzable analog of dGTP and template T or in a wobble conformation when a dG:T mismatch is located at the primer terminus[48]. Similarly, the *Bacillus* DNA polymerase I large fragment active site accommodates a dC:A mismatch in a WC-like or wobble conformation depending on which catalytic metal ion is present[49,50]. For polβ, the largest distortion has been reported for A:C mismatch, where O3′ of the primer terminus sugar is positioned away from the active site via preclusion of direct template base interactions[51–56]. Furthermore, the conformation of G:T mismatch at the primer terminus of the polβ active site is changed between the binary complex and the ternary complex with a correct incoming dNTP. In the binary complex, the mismatch assumes a wobble conformation; however, when the correct incoming dNTP is added, the mismatch changes to a weak Hoogsteen base pair[53–55]. In addition, it has been shown in the structures of polμ that a G:T mismatch adopts a WC-like base pair and that mutations at the polymerase active site cause a change in conformation[57]. Despite the evidence regarding these extensive DNA polymerase structures with a variety of mismatches[48–61], how human DNA ligases engage with various toxic DNA repair intermediates that mimic the polymerase-mediated mutagenic mismatch insertion products at the final ligation step of almost all DNA repair pathways is still unclear.

In the present study, our results revealed that LIG1 counteracts the polβ-promoted mutagenesis products distinctly depending on the architecture of the nick repair intermediate with mismatched ends (Supplementary Fig. 14). Our LIG1/nick DNA complex structures demonstrate that the ligase active site can accommodate a G:T mismatch in the wobble conformation and that the enzyme can fulfill the mutagenic ligation of polβ G:T misinsertion product. On the other hand, LIG1 discriminates against A:C mismatch with a large distortion of 3′-OH and 5′-P ends at the nick. In this case, the nick repair intermediate with the A:C mismatch could serve as a fidelity checkpoint for the removal of the 3′-mismatched base by a proofreading enzyme such as APE1. Accordingly, it has been shown in APE1 structural studies that the nick DNA and instability of mismatched bases facilitate 3′-end cleaning, where a mismatched end is stabilized by protein contacts[38].

Metal ion plays important roles in the ligation reaction during step 1 (deprotonates the lysine nucleophile and activate it to attack on the ATP α-phosphate) and step 3 (activates 3′-OH nucleophilic attack on the 5′-P) of the ligation reaction[62]. DNA ligase/ATP and ligase/ATP/$Mg^{2+}$ complexes for ATP-dependent ligases from other sources, such as *Saccharomyces cerevisiae*, *Pyrococcus furiosus*, *Mycobacterium tuberculosis* LigD, and DNA ligase in a noncovalent complex with AMP, highlight the requirement of metal ions for ligase adenylation[63–72]. The

previously solved crystal structures of LIG1 reveal a $Mg^{2+}$-dependent high-fidelity ($Mg^{HiFi}$) site that is coordinated by the two conserved glutamate residues at the junction between the adenylation (Glu[E]346) and DNA-binding (Glu[E]592) domains of the ligase in direct interaction with DNA[30]. In our study, we used the LIG1 mutant (E346A/E592A or EE/AA) with mutagenesis at the $Mg^{HiFi}$ site for crystallization of LIG1/nick with G:T and A:C mismatches in the absence of $Mg^{2+}$. We compared the RMSD values among our LIG1 EE/AA structures and these previously solved LIG1 structures for cognate C:G and damaged 8-oxoG:A base pairs (Supplementary Table 2). There was no significant difference in either cognate base pair (A:T and C:G) in the presence or absence of the metal ion under crystal conditions for either wild-type or EE/AA mutant of LIG1 (Supplementary Fig. 3a). Furthermore, the superimposition of the LIG1/8-oxoG:A structure with our LIG1/G:T structure showed structural similarity in both structures for the formation of the DNA-AMP intermediate, referring to step 2 of the ligation reaction, in the absence of $Mg^{2+}$ (Supplementary Fig. 15).

The present study represents the human LIG1 structure with G:T and A:C mismatches, particularly at the ligase active site bound to ATP (LIG1-AMP) in step 1 of the ligation reaction, and the importance of the F872 side chain that exhibits a clear shift for positioning the enzyme active site to deter the nick sealing of potentially toxic repair intermediates. As shown for a myriad of DNA polymerases[48–61], our LIG1/G:T mismatch structure reveals the molecular mechanism by which a human ligase can escape mismatch discrimination, leading to the formation of pre-mutagenic repair products. Further structure/function studies with both BER DNA ligases (I and IIIα) are required for all other possible noncanonical base pairs at the 3′-end of the nick DNA to comprehensively understand how human DNA ligases discriminate against the mutagenic repair intermediates containing mismatched or damaged 3′-end which could be formed due to the aberrant gap filling by polβ, which has been reported to be increased in the cancer-associated variants with reduced fidelity[15,16].

## Methods

**Preparation of DNA substrates for crystallization and BER assays.** Oligodeoxyribonucleotides with and without a 6-carboxyfluorescein (FAM) label were obtained from Integrated DNA Technologies (IDT). The nick DNA substrates containing 3′-preinserted correct (dA:T) or mismatches (dG:T and dA:C) with a FAM label at the 3′-end were used for DNA ligation assays in the reaction mixture including LIG1 (wild-type or EE/AA mutant) alone (Supplementary Table 3). The one nucleotide gap DNA substrates containing FAM labels at both 3′- and 5′-ends were used in the coupled assays to observe the ligation of polβ correct or mismatch nucleotide insertion products by LIG1 (wild-type or EE/AA mutant) in the reaction mixture including both polβ and LIG1 (Supplementary Table 4). The nick DNA substrates containing 3′-preinserted correct (dA:T) or mismatches (dG:T and dA:C) with a FAM label at the 5′-end were used for APE1 exonuclease assays in the reaction mixture including APE1 alone (Supplementary Table 5). The nick mismatch containing DNA substrates with FAM labels at both 3′- and 5′-ends were used in the coupled assays to observe APE1 mismatch removal and ligation in the reaction mixture including both APE1 and LIG1 (Supplementary Table 6). The nick DNA substrates containing cognate A:T and mismatches G:T and A:C base pairs were used in the LIG1 X-ray crystallography studies (Supplementary Table 7 and Fig. 1d). All double-stranded DNA substrates were prepared by annealing upstream, downstream, and template primers[17–28].

**Protein purifications.** Human his-tag full-length (1-918) wild-type LIG1 and C-terminal (Δ261) E346A/E592A (EE/AA) LIG1 mutant were overexpressed in Rosetta (DE3) pLysS *E. coli* cells (Millipore Sigma) and grown in Terrific Broth (TB) media with kanamycin (50 μg ml⁻¹) and chloramphenicol (34 μg ml⁻¹) at 37 °C[17–28]. Once the OD was reached 1.0, the cells were induced with 0.5 mM isopropyl β-D-thiogalactoside (IPTG) and the overexpression was continued overnight at 28 °C. After the centrifugation, the cells were lysed in the lysis buffer containing 50 mM Tris-HCl (pH 7.0), 500 mM NaCl, 20 mM imidazole, 10% glycerol, 1 mM PMSF, an EDTA-free protease inhibitor cocktail tablet by sonication at 4 °C. The lysate was pelleted at 31,000 x g for 1 h at 4 °C. The cell lysis solution was filter clarified and then loaded onto the HisTrap HP column (GE

Health Sciences) that was previously equilibrated with the binding buffer including 50 mM Tris-HCl (pH 7.0), 500 mM NaCl, 20 mM imidazole, and 10% glycerol. The column was washed with the binding buffer and then followed by washing buffer containing 50 mM Tris-HCl (pH 7.0), 500 mM NaCl, 35 mM imidazole, and 10% glycerol. The protein was finally eluted with an increasing imidazole gradient 0–500 mM at 4 °C. The collected fractions were then subsequently loaded onto HiTrap Heparin (GE Health Sciences) column that was equilibrated with the binding buffer containing 50 mM Tris-HCl (pH 7.0), 50 mM NaCl, 1 mM EDTA, 5% glycerol, and the protein was eluted with a linear gradient of NaCl up to 1 M. LIG1 protein was further purified by Resource Q and finally by Superdex 200 10/300 (GE Health Sciences) columns in the buffer containing 20 mM Tris-HCl (pH 7.0), 200 mM NaCl, and 1 mM DTT.

Human his-tag full-length APE1 was overexpressed in BL21(DE3) E. coli cells (Invitrogen) in Lysogeny Broth (LB) media at 37 °C for 8 h, induced with 0.5 mM IPTG and the overexpression was continued for overnight at 28 °C. After the cells were harvested, lysed at 4 °C, and then clarified as described above, the supernatant was loaded onto HisTrap HP column (GE Health Sciences) and purified with an increasing imidazole gradient (0-300 mM) elution at 4 °C. The collected fractions were then subsequently loaded onto HiTrap Heparin column (GE Health Sciences) with a linear gradient of NaCl up to 1 M. APE1 protein was then further purified by Superdex 200 increase 10/300 chromatography (GE Healthcare) in the buffer containing 20 mM Tris-HCl (pH 7.0), 200 mM NaCl, and 1 mM DTT.

Human GST-tag full-length polβ was overexpressed in One Shot BL21(DE3) E. coli cells (Invitrogen) in LB media at 37 °C for 8 h, induced with 0.5 mM IPTG, and the overexpression was continued for overnight at 28 °C[17–28]. After cell lysis at 4 °C by sonication in the lysis buffer containing 1X PBS (pH 7.3), 200 mM NaCl, 1 mM DTT, cOmplete Protease Inhibitor Cocktail (Roche), the lysate was pelleted at 31,000 x g for 1 h and then clarified by centrifugation and filtration. The supernatant was loaded onto GSTrap HP column (GE Health Sciences) and purified with the elution buffer containing 50 mM Tris-HCl (pH 8.0) and 10 mM reduced glutathione. To cleave a GST-tag, the recombinant protein was incubated with PreScission Protease (GE Health Sciences) for 16 h at 4 °C in the buffer containing 1X PBS (pH 7.3), 200 mM NaCl, and 1 mM DTT. After the cleavage, the polβ protein was subsequently passed through a GSTrap HP column, and the protein without GST-tag was then further purified by loading onto Superdex 200 gel filtration column (GE Health Sciences) in the buffer containing 50 mM Tris-HCl (pH 7.5) and 400 mM NaCl. All proteins purified in this study were dialyzed against the storage buffer including 25 mM Tris-HCl (pH 7.0), 200 mM NaCl, concentrated, frozen in liquid nitrogen, and stored at −80 °C. Protein quality was evaluated onto 10% SDS-PAGE, and the protein concentration was measured using absorbance at 280 nm.

**Crystallization and structure determination**. LIG1 C-terminal (Δ261) EE/AA mutant was used for crystals production. All LIG1-DNA complex crystals were grown at 20 °C using the hanging drop method. LIG1 (at 26 mgml⁻¹)/DNA complex solution was prepared in 20 mM Tris-HCl (pH 7.0), 200 mM NaCl, 1 mM DTT, 1 mM EDTA and 1 mM ATP at 1.5:1 DNA:protein molar ratio and then mixed with 1 μl reservoir solution containing 100 mM MES (pH 6.6), 100 mM lithium acetate, and 20% (w/v) polyethylene glycol PEG3350. All crystals grew in 1–2 days. They were then washed in the reservoir solution with 20% glycerol and flash cooled in liquid nitrogen for data collection. The crystals were maintained at 100 K during X-ray diffraction data collection using the beamline 7B2 at Cornell High Energy Synchrotron Source (CHESS). The diffraction images were indexed and integrated using HKL2000. All structures were solved by the molecular replacement method using PHASER using PDB entry 6P0D as a search model[73,74]. Iterative rounds of model building in COOT and refinement with PHENIX or REFMAC5 were used to produce the final models[75–77]. 3D program was used for sugar pucker analysis[78]. All structural images were drawn using PyMOL (The PyMOL Molecular Graphics System, V0.99, Schrödinger, LLC). Detailed crystallographic statistics are provided in Table 1.

**DNA ligation assays**. The ligation assays using the nick DNA substrates containing 3′-preinserted correct dA:T, mismatches dG:T and dA:C were performed to test the ligation efficiency of wild-type and EE/AA mutant of LIG1 (Supplementary Fig. 5a)[17–28]. Briefly, the ligation assays were performed in the mixture containing 50 mM Tris-HCl (pH 7.5), 100 mM KCl, 10 mM MgCl₂, 1 mM ATP, 1 mM DTT, 100 μgml⁻¹ BSA, 10% glycerol, and 500 nM DNA substrate in a final volume of 10 μl. The reactions were initiated by the addition of 100 nM LIG1 (wild-type or EE/AA mutant), incubated at 37 °C, and stopped at the time points indicated in the figure legends. The reaction products were then quenched with an equal amount of gel loading buffer containing 95% formamide, 20 mM ethylenediaminetetraacetic acid, 0.02% bromophenol blue and 0.02% xylene cyanol. After incubation at 95 °C for 3 min, the reaction products were separated by electrophoresis on an 18% denaturing polyacrylamide gel. The gels were scanned with a Typhoon PhosphorImager (Amersham Typhoon RGB), and the data were analyzed using ImageQuant software.

**BER assays to measure DNA ligation of polβ nucleotide insertion products**. The coupled assays using the one nucleotide gap DNA substrates with template

A or C were performed to test the ligation of polβ nucleotide insertion (correct or mismatch) in the reaction mixture including both polβ and LIG1 (Supplementary Fig. 5b)[17–28]. Briefly, the coupled assays were performed in a mixture containing 50 mM Tris-HCl (pH 7.5), 100 mM KCl, 10 mM MgCl₂, 1 mM ATP, 1 mM DTT, 100 μgml⁻¹ BSA, 10% glycerol, 100 μM dNTP, and 500 nM DNA substrate in a final volume of 10 μl. The reactions were initiated by the addition of pre-incubated enzyme mixture of polβ/LIG1 (100 nM) and incubated at 37 °C for the time points as indicated in the figure legends. The reaction products were then mixed with an equal amount of gel loading buffer, separated by electrophoresis on an 18% denaturing polyacrylamide gel, and analyzed as described above.

**BER assays to measure APE1 exonuclease activity and LIG1 ligation**. APE1 exonuclease assays using the nick DNA substrates with 3′-preinserted mismatches dG:T and dA:C were performed to examine APE1 proofreading role for removing a mismatched base (Supplementary Fig. 11a). Briefly, APE1 activity assays were performed in the reaction mixture containing 50 mM HEPES (pH 7.4), 100 mM KCl, 3 mM MgCl₂, 0.1 mg ml⁻¹ BSA, and 500 nM DNA substrate in a final volume of 10 μl. The reactions were initiated by the addition of 50 nM APE1, incubated at 37 °C for the time points as indicated in the figure legends, quenched by mixing with 100 mM EDTA, and then mixed with an equal amount of gel loading buffer. The nick DNA substrates including 3′-preinserted mismatches dG:T and dA:C were used for repair assays to test APE1 exonuclease and DNA ligation activities in the same reaction mixture (Supplementary Fig. 11b). Briefly, the repair assays were performed in a mixture containing 50 mM HEPES (pH 7.4), 100 mM KCl, 5 mM MgCl₂, 1 mM ATP, 0.1 mg ml⁻¹ BSA, and 500 nM DNA substrate in a final volume of 10 μl. The reactions were initiated by the addition of pre-incubated enzyme mixture including APE1/LIG1 (100 nM), incubated at 37 °C for the time points as indicated in the figure legends. The reaction products were quenched by mixing with 100 mM EDTA and then mixed with an equal amount of gel loading buffer. The reaction products were separated by electrophoresis on an 18% denaturing polyacrylamide gel and analyzed as described above.

**APE1 and LIG1 protein-protein interaction assay**. The protein-protein interaction between APE1 and LIG1 was measured by surface plasmon resonance (SPR) in real time using Biacore X-100 (GE Healthcare)[28]. Briefly, one flow cell of the CM5 sensor chip was activated at 25 °C with 1:1 mixture of 0.2 M EDC and 0.05 M NHS in water, and then APE1 protein was injected over the flow cell in 10 mM sodium acetate at pH 5.0 at a flow rate of 10 μl/min. The binding sites were blocked using 1 M ethanolamine. LIG1 (at the concentration range of 0–1.6 μM) was then injected for 3 min at a flow rate of 30 μl/min in the binding buffer containing 20 mM HEPES pH 7.4, 150 mM NaCl, 3 mM EDTA and 0.005% (v/v) Surfactant P20. After a dissociation phase for 3–4 min, 0.2% SDS was injected for 30 sec to regenerate the chip surface. Non-specific binding to a blank flow cell was subtracted to obtain corrected sensorgrams. All data were analyzed using BIAevaluation software version 2.0.1 and fitted to a 1:1 (Langmuir) binding model to obtain equilibrium constant (KD).

**Reporting summary**. Further information on research design is available in the Nature Research Reporting Summary linked to this article.

## Data availability

The data that support this study are available from the corresponding author upon request. Atomic coordinates and structure factors for the reported crystal structures have been deposited in the RCSB Protein Data Bank under accession numbers 7SUM, 7SXE, and 7SX5. Source data are provided with this paper.

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

## Acknowledgements

This work is based upon research conducted at the Center for High Energy X-ray Sciences (CHEXS), which is supported by the National Science Foundation under award DMR−1829070, and the Macromolecular Diffraction at CHESS (MacCHESS) facility, which is supported by award 1-P30-GM124166-01A1 from the National Institute of General Medical Sciences, National Institutes of Health, and by New York State's Empire State Development Corporation (NYSTAR). The authors thank Jacob E. Combs (McKenna Lab, University of Florida) for his assistance with crystal shipment and data collection. This work was supported by the National Institutes of Health/National Institute of Environmental Health Sciences Grant 4R00ES026191 and the University of Florida Thomas H. Maren Junior Investigator Fund P0158597 to M.Ç.

## Author contributions

Conceptualization M.Ç., methodology and investigation M.Ç., M.G., T.Q.; writing-original draft, M.Ç., T.Q.; writing-reviewing, editing original draft and revision, M.Ç., T.Q., M.G., and R.M.; funding acquisition M.Ç.

## Competing interests

The authors declare no competing interests.
