## [Peer Review File · Nature Communications]

Structures of LIG1 that engage with mutagenic mismatches inserted by pol β in base excision repairReviewers' Comments:

Reviewer #1:

Remarks to the Author:

The manuscript from Caglayan's group titled "Structures of LIG1 engaging with mutagenic mismatches inserted by pol β in base excision repair" describes structures of human Ligase 1 with nicked DNA duplexes containing either cognate A:T or mismatched G:T or A:C pairs. The authors demonstrate that in the A:T and G:T structures LIG1 proceeds to step 2 of the reaction and the 5'-phosphate of the nicked strand is adenylated. However, in the A:C structure, LIG1 is stuck in the step 1 with the conserved lysine residue covalently adenylated. The biochemical data provided in the manuscript support the structural findings and show efficient sealing of the DNA with G:T but not with A:C mismatch. These are important findings highlighting that LIG1 can modulate the fidelity of BER and is able to seal nicks in G:T-mismatched duplexes, an error often generated by BER polymerase β .

Despite the interesting findings the manuscript is very hard to follow and should be considerably re-written.

For instance, the Abstract has the following sentence:

"Our structures revealed that LIG1 active site can accommodate G:T mismatch in a similar conformation with A:T base pairing, while it stays in the LIG1-adenylate intermediate during initial step of ligation reaction in the presence of A:C mismatch at 3'-strand."

The reader might think that LIG1 accommodates the G:T mismatch with Watson-Crick-like geometry similarly to cognate A:T pair (like it was observed in Pol λ , ref 50). Instead, only later in the manuscript one learns that the G:T mismatch assumes expected wobble configuration.

The Introduction is way too long and most importantly lacks clarity.

Statement on page 4, line 84 is misleading:

"Furthermore, it has been reported that Watson-Crick like G:T mismatch, if left unrepaired, could lead..."

The G:T mismatch in DNA normally adopts a wobble configuration with a thymine base shifted into the major groove. The dGTP:T mismatch with Watson-Crick-like geometry in a rare ionized form was indeed observed in the active site of Pol λ , ref 50). However, upon covalent incorporation of G and further extension of the growing primer strand, the G:T mismatch will likely be present in the wobble conformation (see for example Johnson and Beese, Cell 2004). Similarly, a wobble G:T mismatch was found in a binary Pol β -DNA complex ref 64.

Page 4, line 82:

"For example, G:T mismatches are among the more prevalent mismatches found in nature".

Is it "in nature" or in a human genome?

Other points:

1. The overall structure of the LIG1-DNA complex should be briefly described in the Results and Supplementary Figure 1 illustrating it should be moved to Figure 1 as a panel.
2. Figure 1 should display omit difference maps for AMP (either simulated annealing or composite), not 2Fo-Fc that could be model-biased especially at $\sim 3\text{\AA}$ resolution.
3. Same point applies to Supplementary Figure 3, displaying the maps for the A:T and the mismatched base pairs. These maps should be better shown in Figure 2A, since it shows these pairs and Supplementary Figure 3 could be eliminated. Also, please show C1'-C1' distances for the pairs.
4. Figure 4 compares the differences in placement of the wobble A:C mismatch pair and the cognate A:T pair in the active site of LIG1. The authors note the shift in the position of Phe 872.

What happens to the other conserved residue of the OB-fold domain, Phe 635? Both residues, Phe 635 and Phe 872, are inserted into the minor groove of the cognate DNA duplex (Nature 2004, 432).

Did the orientation of the OB-fold domain changed? The authors should superimpose the structures by the DNA-binding (DBD) and Adenylation domain (AdD), separately and together, to evaluate if OBD is positioned differently in the A:C structure.

5. Supplementary Figure 1:

"Schematic view of LIG1 domain composition including the N-terminal domain (amino acids 1-261, gray),"

Residues 1-261 do not form a domain, they should be referred as the N-terminal region. This region contains a nuclear localization signal as well as residues that interact with other proteins (polymerase β and PCNA)

"and the catalytic core (amino acids 262-919) consisting of the DNA-binding domain (DBD, pink), Adenylation domain (AdD, yellow), and OB-fold domain (OBD, green)."

DNA-binding domain (DBD) is not a part of the catalytic core, see ref.3.

6. The Results section has many descriptions of the previously published works. These should be in Discussion.

Reviewer #2:

Remarks to the Author:

The goal of the authors is to understand the fidelity of the base-excision repair pathway, focusing on the role of human DNA ligase 1 (LIG1).

The manuscript reports three crystal structures (2.8 - 3.0 Å resolution) of LIG1 bound to DNA substrates containing a nick and either a correct basepair (A:T) or different mismatches (G:T and A:C) on the 3' side of the nick. The protein contained mutations of E346A and E592A (EE/AA) at a Mg²⁺ binding site that was designated the high fidelity (HiFi) site by others (ref. 54). The A:T and G:T structures were captured at the second step of the ligation reaction, with a 5'-DNA-AMP intermediate, while the A:C structure was captured at the first step of the reactions, with LIG1-adenylate intermediate.

When the three structures are compared, the conformation of the nucleotide at the 3' side of the nick differs in the mispaired structures compared the the correct A:T pair. The most significant difference is for the A:C mispair, in which the 3'OH is rotated by 50° from its position in the A:T structure. The DNA on the 5' side of the nick in the A:C structure is also displace relative to the other structures.

Biochemical data shows that substrates with a 3'-dA:C mispair are not efficiently ligated compared to 3'dA:T and 3'dG-T substrates, supporting the proposal that the conformational differences in the structure allows the LIG1 to avoid performing mutagenic ligation with the A:C mispair but not the G:T mispair. Data is also provided showing that AP-Endonuclease 1 (APE1) interacts with LIG1 and can remove the 3' mismatched base in both the A:C and G:T mispairs.

The data presented provide important insights into the contributions of LIG1 to the fidelity of the base-excision repair pathway. It is notable that the A:C structure is the only one available showing an intermediate at step 1 of the ligation reaction.

Some additional information, analysis and discussion would enhance the impact of the manuscript.

Some key questions:

- If Mg²⁺ is required for the ligation reaction, how were the structures obtained when Mg²⁺ was not included in the purification, crystallization or cryoprotection solutions? Were the reaction intermediates produced because trace Mg²⁺ in the reagents?

- How do these structures relate to those reported in reference 54? In particular, how might the EE/AA mutation and the absence of Mg²⁺ alter the conformation of the DNA and/or protein and impact the interpretation of the data presented here?

Additional questions & comments:

1. Lines 159-160, Figure 2C: What is the center of rotation for the 3'OH in the A:C structure? Does the ribose adopt a different sugar pucker? What is the distance between the 3'OH and where the 5'P is located in the A:T and G:T structures? Could the different conformation be due to the complex being at step 1 vs 2 of the reaction?

2. Lines 160-165, Figure 3: The positions of nucleotides in the upstream DNA only seem to be shifted in the A:C structure (Fig. 3A), not in the G:T structure (Fig. 3B), but the text indicates that there are shifts in both structures.

3. Lines 186-188. This sentence is confusing. Is the implication that the absence of Mg²⁺ and E346 and E592 in the structures reported here does not alter the structure compared to the G:C structure? What is the RMSD among the structures?

4. Line 217: Is "dGTP:C" a typo? The sentence refers to it as a mismatch. Should it be "dGTP:T" instead?

5. Line 220: "In consistent with our previous studies..." does not make sense. Should it be "Inconsistent" or "Consistent"?

6. Lines 250-251, Figure 7: In the coupled ligation reactions, it appears that pol beta does not insert dA opposite C so that it is not possible to conclude whether or not LIG1 was able to ligate that product (although even when that product is preformed in Figure 6, the ligation reaction is very inefficient, suggesting that both steps are defective). The labels in Fig. 7B are misaligned. It would also be helpful to have a diagram explaining this experiment.

7. The manuscript would benefit from careful proofreading. There are small mistakes throughout.

May 4th, 2022

Ref: NCOMMS-21-48112

Structures of LIG1 that engage with mutagenic mismatches inserted by pol β in base excision repair

Point-by-point responses to the Referees' comments are as follows:

Reviewer #1:

The manuscript from Caglayan's group titled "Structures of LIG1 engaging with mutagenic mismatches inserted by pol β in base excision repair" describes structures of human Ligase 1 with nicked DNA duplexes containing either cognate A:T or mismatched G:T or A:C pairs. The authors demonstrate that in the A:T and G:T structures LIG1 proceeds to step 2 of the reaction and the 5'-phosphate of the nicked strand is adenylated. However, in the A:C structure, LIG1 is stuck in the step 1 with the conserved lysine residue covalently adenylated. The biochemical data provided in the manuscript support the structural findings and show efficient sealing of the DNA with G:T but not with A:C mismatch. These are important findings highlighting that LIG1 can modulate the fidelity of BER and is able to seal nicks in G:T-mismatched duplexes, an error often generated by BER polymerase β . Despite the interesting findings the manuscript is very hard to follow and should be considerably re-written.

Response: We thank the Referee for these helpful comments. We substantially revised the manuscript to address the concerns and the Referee's points as summarized below. In order to accommodate the request for editing of the English text, we worked with professional editing service, Springer Nature Author Service, for English language editing. The revised manuscript was significantly edited for grammar, phrasing, and punctuation to improve the clarity, flow and readability of the text.

Point 2: For instance, the Abstract has the following sentence:

"Our structures revealed that LIG1 active site can accommodate G:T mismatch in a similar conformation with A:T base pairing, while it stays in the LIG1-adenylate intermediate during initial step of ligation reaction in the presence of A:C mismatch at 3'-strand." The reader might think that LIG1 accommodates the G:T mismatch with Watson-Crick-like geometry similarly to cognate A:T pair (like it was observed in Pol λ , ref 50). Instead, only later in the manuscript one learns that the G:T mismatch assumes expected wobble configuration.

Response: We thank the Reviewer for pointing out this inconsistency. We corrected the abstract as following: *"Here, we determined the X-ray structures of LIG1/nick DNA complexes with G:T and A:C mismatches and uncovered the ligase strategies that favor or deter the ligation of base substitution errors. Our structures revealed that the LIG1 active site can accommodate a G:T mismatch in the wobble conformation, where an adenylate (AMP) is transferred to the 5'-phosphate of a nick (DNA-AMP), while it stays in the LIG1-AMP intermediate during the initial step of the ligation reaction in the presence of an A:C mismatch at the 3'-strand."*

Point 3: The Introduction is way too long and most importantly lacks clarity.

Response: We considerably shortened the introduction part and moved all literature summaries about previously solved LIG1 structures to the discussion part of the revised manuscript. Introduction part of the revised manuscript only focuses on the significance of the study after a

brief background about DNA ligase I and ligation reaction at the downstream steps of base excision repair.

Point 4: Statement on page 4, line 84 is misleading:

“Furthermore, it has been reported that Watson-Crick like G:T mismatch, if left unrepaired, could lead...”

Response: Thank the Reviewer for carefully reading the manuscript.

Elegant structure and kinetics studies have provided the evidence showing that wobble and Watson-crick like mismatches can exist within polymerase and ribosome active sites, strongly suggest that energetic competition between wobble and Watson-crick like mismatches is robust and is a key determinant of misincorporation probability during replication and translation in addition to their proposed roles in the transcription and DNA damage repair.

To address the Reviewer’s point here, we corrected the statement in the discussion part of the revised manuscript as following: “*Spontaneous mutagenesis from DNA replication errors has been a prominent source of base substitution errors in the tumor suppressor genes in different types of cancer and the intrinsic formation of Watson-crick like mismatches has been reported as an important determinant of the base substitution mutagenesis*”.

Point 5: The G:T mismatch in DNA normally adopts a wobble configuration with a thymine base shifted into the major groove. The dGTP:T mismatch with Watson-Crick-like geometry in a rare ionized form was indeed observed in the active site of Pol λ , ref 50). However, upon covalent incorporation of G and further extension of the growing primer strand, the G:T mismatch will likely be present in the wobble conformation (see for example Johnson and Beese, Cell 2004). Similarly, a wobble G:T mismatch was found in a binary Pol β -DNA complex ref 64.

Response: Thank the Reviewer for this very helpful information. The mechanisms of mismatch discrimination by DNA polymerases have been extensively reported through X-ray structural studies. Our LIG1 structures represent the first study for a human DNA ligase with A:C and G:T mismatches at the ligase active site in two different steps of the ligation reaction (step 1 and 2, respectively). In the discussion part of the revised manuscript, we added a section describing previously published studies for DNA polymerases in wobble versus Watson-Crick-like mismatches, particularly dG:T, dT:G, dA:C, and dC:A, reported for human X family DNA polymerases pol β , pol λ , pol μ , and *Bacillus stearothermophilus* DNA polymerase I large fragment (BF). Also, in the revised manuscript, we added Supplementary Table 1 showing the DNA polymerase/mismatch structures from diverse families and organisms to compare the differences in the conformation of a particular mismatch depending on its location at the polymerase active site, the absence versus presence of a catalytic metal ion, and electrostatic environment of the active site.

Point 6:

Page 4, line 82: “For example, G:T mismatches are among the more prevalent mismatches found in nature”. Is it “in nature” or in a human genome?

Response:

Watson and Crick proposed that spontaneous base substitutions could be a consequence of bases spontaneously pairing in rare tautomeric forms and two possible transition mispairs, G:T and A:C, involve the enol form of guanine or thymine and the imino form of adenine or cytosine, respectively (Watson, J.D. and Crick, F.H., Nature, 171, 954-967, 1953). Both mismatches fit well within the dimensions of the DNA double helix to preserve the geometry of a correct Watson-Crick base pair respectively (Watson, J.D. and Crick, F.H., Nature, 171, 737-738, 1953). In the present study, we solved the LIG1 structures for these two particular mismatches: G:T and A:C.

The effects of the A-C mismatch on helix stability and dynamics differ from those of the G-T pair. Furthermore, it is very likely that parameters such as polymerase types, DNA sequence, and the

chemical environment can affect the relative stabilities and lifetimes of the anionic, tautomeric, and wobble versus Watson-crick like conformations, which may help to explain the broad range of misincorporation probabilities as we summarized in our response to Point 5 above.

In order to address the Referee's point here, we added a section to the discussion part of the revised manuscript as following: "*The G:T and A:C mismatches suit well within the dimensions of the DNA double helix to maintain the geometry of a Watson-Crick base pair and the effects of the A:C mismatch on helix stability and dynamics differ from those of the G:T base pair. The G:T is the most common mismatch arising from the deamination of 5-methylcytosine to thymine and the conformation and dynamics of G:T mismatch (wobble versus Watson-Crick) are the best studied base pair by X-ray crystallography and NMR relaxation dispersion studies in DNA and RNA duplexes. Our study represent the LIG1 structures for these particular G:T and A:C mismatches at the ligase active site*".

Point 7:

The overall structure of the LIG1-DNA complex should be briefly described in the Results and Supplementary Figure 1 illustrating it should be moved to Figure 1 as a panel.

Response: In order to address the Referee's point, we moved Supplementary Figure 1 showing the schematic view of the nick DNA substrate used in LIG1/mismatch crystals to Figure 1D in the revised manuscript.

Also, we added a statement summarizing our structures (Figures 1-6)

In the results section "*Overall, our LIG1/mismatch structures demonstrate that the ligase is trapped as the adenylated-DNA intermediate (AMP-DNA), which favors the mutagenic ligation of the G:T mismatch, and the active site remains in an inactive conformation (AMP-LIG1), which deters nick sealing of the A:C mismatch*"

In the discussion section describing our working model: "*In the present study, our results revealed that LIG1 counteracts the pol β -promoted mutagenesis products distinctly depending on the architecture of the nick repair intermediate with mismatched ends (Supplementary Scheme 3). Our LIG1/nick DNA complex structures demonstrated that the ligase active site can accommodate a G:T mismatch in the wobble conformation and that the enzyme can fulfill the mutagenic ligation of pol β G:T misinsertion product. On the other hand, LIG1 discriminates against A:C mismatch with a large distortion of 3'-OH and 5'-P ends at the nick. In this case, the nick repair intermediate with the A:C mismatch could serve as a fidelity checkpoint for the removal of the 3'-mismatched base by a proofreading enzyme such as APE1. Accordingly, it has been shown in APE1 structural studies that the nick DNA and instability of mismatched bases facilitate 3'-end cleaning, where a mismatched end is stabilized by protein contacts*".

Point 8:

Figure 1 should display omit difference maps for AMP (either simulated annealing or composite), not 2Fo-Fc that could be model-biased especially at ~3Å resolution.

Response: For all LIG1 structures (A:T, G:T, and A:C), we prepared a new figure showing the omit difference maps for AMP. We replaced Figure 1 of the original manuscript with this display in the revised manuscript (Figure 1).

Point 9:

Same point applies to Supplementary Figure 3, displaying the maps for the A:T and the mismatched base pairs. These maps should be better shown in Figure 2A, since it shows these pairs and Supplementary Figure 3 could be eliminated. Also, please show C1'-C1' distances for the pairs.

Response: In order to address the Referee's point, we removed Supplementary Figure 3. In the Figure 2 of the revised manuscript, we presented C1'-C1' distances for A:T, G:T and A:C pairs.

Point 10:

Figure 4 compares the differences in placement of the wobble A:C mismatch pair and the cognate A:T pair in the active site of LIG1. The authors note the shift in the position of Phe 872.

Response:

As reported in the first LIG1 structure (Pascal, J. M. *et al. Nature*, 473-478, 2004), the OB-fold domain (OBD) interacts with the minor groove of the DNA and the contacts between Adenylation domain (AdD) and OBD are critical for correctly positioning these domains when they engage with a nick DNA. During catalysis, the AdD forms a salt bridge with the OBD via Asp570 and Arg871. This positioning of the AdD and OBD domains creates a surface that binds to DNA and such interaction stabilizes the catalytic domains in a conformation where they fully encircle a nick. Notably, Phe(F)635 and Phe(F)872 that are located on the AdD and OBD, respectively, are forced into the minor groove both 3' and 5' ends of the nick. As a result of these interactions, the DNA duplex upstream of the nick duplex assumes an A form structure and the nick is opened for ligation.

In our study, we showed that the position of F872 exhibits an important difference between the LIG1 structures for cognate A:T and mismatch A:C (Figure 4). The overlay of both LIG1 structures demonstrated that the F872 distorts the alignment upstream of the A:T nick, where -1G and +1A nucleotides are in parallel between the 3' and 5' strands. To address the referee's point here, we noted this finding in the revised manuscript as following: "*the superimposition of the structures in placement of the wobble A:C mismatch and the cognate A:T base pair at the active site of LIG1 demonstrated the shift in the position of F872*".

Point 11:

What happens to the other conserved residue of the OB-fold domain, Phe 635? Both residues, Phe 635 and Phe 872, are inserted into the minor groove of the cognate DNA duplex (Nature 2004, 432).

Response:

As reported in the first LIG1 structure (Pascal, J. M. *et al. Nature*, 473-478, 2004), F635 residue that is situated near the upstream end where it is positioned near the sugar moiety of the 3'-end plays a different role in catalysis. Furthermore, the importance of F635 residue have been shown in the Chlorella virus DNA ligase - ChVLig (Samai, P. & Shuman, S., *J. Biol. Chem.*, 2011). In this study, the effects of 35 mutations residing in the NTase domain have been characterized on ChVLig activity *in vivo* and *in vitro*, using *cdc9* (homolog of human DNA ligase I)-deficient yeast cells. The authors demonstrated that the mutations cause a defect in the ability of ChVLig to complement LIG1 deficiency in yeast cells.

To address the referee's point here, we evaluated the positions of F872 and F635 in the LIG1 A:T and A:C structures and the superimposition of both structures does not show any conformational difference at the F635 side chain. We presented it as Supplementary Figure 4 in the revised manuscript. Although it's out of scope for this present study, as a follow-up work, we generated Phe to Ala mutations for F635 and F872 residues to further investigate the importance of these side chains for the substrate discrimination mechanism of LIG1 against the nick DNA containing mismatched or damaged base pairs at 3'-end.

Point 12:

Did the orientation of the OB-fold domain changed? The authors should superimpose the structures by the DNA-binding (DBD) and Adenylation domain (AdD), separately and together, to evaluate if OBD is positioned differently in the A:C structure.

Response: In order to address the referee's point here, we superimposed LIG1 A:T and A:C structures that shows the OB-fold (OBD), DNA-binding (DBD) and Adenylation (AdD) domains together. This overlay demonstrates a loop region in the DBD domain corresponding to amino

acid residues 388-395, which is positioned differently in the A:C structure. We present this new overlay as Supplementary Figure 3B along with the superimposition of our all A:T, G:T, A:C structures (LIG1 EE/AA mutant) and previously reported C:G structure (wild-type LIG1) in the revised manuscript (Supplementary Figure 3).

Point 13:

Supplementary Figure 1:

“Schematic view of LIG1 domain composition including the N-terminal domain (amino acids 1-261, gray),” Residues 1-261 do not form a domain, they should be referred as the N-terminal region. This region contains a nuclear localization signal as well as residues that interact with other proteins (polymerase β and PCNA) and the catalytic core (amino acids 262-919) consisting of the DNA-binding domain (DBD, pink), Adenylation domain (AdD, yellow), and OB-fold domain (OBD, green).” DNA-binding domain (DBD) is not a part of the catalytic core, see ref.3.

Response:

We thank the Referee for this point. As we discussed extensively in our review article (Caglayan M., *J. Mol. Biol.* 2019), all three mammalian DNA ligases as well as prototypical bacterial DNA ligases contain a highly conserved C-terminal catalytic core consisting of the oligonucleotide binding domain (OBD) and the adenylation domain (AdD) or nucleotidyl transferase domain. Mammalian DNA ligases also harbor the DNA binding domain (DBD) that stabilizes the DNA and interacts with both the minor groove of the nicked DNA as well as the OBD to make a ring-shaped protein structure that encircles the DNA during nick sealing. This domain is conserved in all ATP-dependent eukaryotic DNA ligases and stimulates the nick joining enzymatic activity of the catalytic core consisting of AdD and OBD domains in several multidomain DNA ligases. Human DNA ligase I protein contains the N-terminal region, which is missing from the crystal structure, and includes the PCNA interacting peptide and a nuclear localization signal. This noncatalytic part of LIG1 mediates its interaction with DNA-sliding clamps and pol β during DNA repair and replication. In order to address the Referee’s point here, we revised the scheme showing the protein organization of human DNA ligase I in Supplementary Figure 2 of the revised manuscript.

Point 14:

The Results section has many descriptions of the previously published works. These should be in Discussion.

Response: We thank the Reviewer for this helpful comment. We removed the parts describing previously published studies from the introduction part. The most important study that is highly relevant to our present LIG1 structures is our first report (Caglayan M. *Nucleic Acids Res.*, 2020) where we demonstrated that the BER DNA ligases, DNA ligase I and DNA ligase III α /XRCC1 complex, are compromised by subtle changes in all 12 possible noncanonical base pairs at the 3'-end of the nicked repair intermediate. We also reported how pol β mismatch nucleotide insertions affects the efficiency of subsequent ligation step in the BER pathway. With these initial findings and our present LIG1/mismatch structures, our overall studies contribute to understanding of how the identity of the mismatch affects the fidelity of BER at the final steps. In addition, in our present LIG1/mismatch structure work, we further extended this picture by investigating the role of APE1 in the removal of mismatched bases with its exonuclease activity from potentially mutagenic repair intermediates.

Reviewer #2:

The goal of the authors is to understand the fidelity of the base-excision repair pathway, focusing on the role of human DNA ligase 1 (LIG1).

The manuscript reports three crystal structures (2.8 - 3.0 Å resolution) of LIG1 bound to DNA substrates containing a nick and either a correct basepair (A:T) or different mismatches (G:T and A:C) on the 3' side of the nick. The protein contained mutations of E346A and E592A (EE/AA) at a Mg²⁺ binding site that was designated the high fidelity (HiFi) site by others (ref. 54). The A:T and G:T structures were captured at the second step of the ligation reaction, with a 5'-DNA-AMP intermediate, while the A:C structure was captured at the first step of the reactions, with LIG1-adenylate intermediate. When the three structures are compared, the conformation of the nucleotide at the 3' side of the nick differs in the mispaired structures compared the the correct A:T pair. The most significant difference is for the A:C mispair, in which the 3'OH is rotated by 50° from its position in the A:T structure. The DNA on the 5' side of the nick in the A:C structure is also displace relative to the other structures.

Biochemical data shows that substrates with a 3'-dA:C mispair are not efficiently ligated compared to 3'-dA:T and 3'-dG-T substrates, supporting the proposal that the conformational differences in the structure allows the LIG1 to avoid performing mutagenic ligation with the A:C mispair but not the G:T mispair. Data is also provided showing that AP-Endonuclease 1 (APE1) interacts with LIG1 and can remove the 3' mismatched base in both the A:C and G:T mispairs. The data presented provide important insights into the contributions of LIG1 to the fidelity of the base-excision repair pathway. It is notable that the A:C structure is the only one available showing an intermediate at step 1 of the ligation reaction. Some additional information, analysis and discussion would enhance the impact of the manuscript.

Some key questions:

Point 1:

If Mg²⁺ is required for the ligation reaction, how were the structures obtained when Mg²⁺ was not included in the purification, crystallization or cryoprotection solutions? Were the reaction intermediates produced because trace Mg²⁺ in the reagents?

Response:

Metal ion plays important roles in the ligation reaction, specifically at step 1 (deprotonates the lysine nucleophile and activate it to attack on the ATP α-phosphate and stabilize the pentavalent transition state formed) and step 3 (activates 3'-OH nucleophilic attack on the 5'-P and form the pentavalent transition) as reported (Unciuleac M.C, Goldgur Y., and Shuman S. *PNAS*, 2017). The previously solved crystal structures of DNA ligase/ATP and ligase/ATP/Mg²⁺ complexes for ATP-dependent ligases from other sources, such as *Saccharomyces cerevisiae*, *Pyrococcus furiosus*, *Mycobacterium tuberculosis* LigD, and DNA ligase in a noncovalent complex with AMP, highlight the requirement of metal ions for ligase adenylation. We added this important information regarding the role of metal ion for the ligation reaction and the ligase structures from other sources to the discussion part of the revised manuscript.

The previously solved structures of LIG1 reveal a Mg²⁺-dependent high fidelity (Mg^{HiFi}) site that is coordinated by the two conserved glutamate residues at the junction between the adenylation (Glu[E]346) and DNA-binding (Glu[E]592) domains of the ligase in direct interaction with DNA. Moreover, mutations at these glutamate residues to alanine (E346A/E592A or EE/AA) lead to an LIG1 enzyme with lower fidelity (Tumbale, P. et al. *Nat Commun.*, 2019). In our recent study (Kamble, P. et al. *J. Biol. Chem.*, 2021), we demonstrated how such a distinct environment, which the staggered LIG1 conformation creates due to mutations at the Mg^{HiFi} metal site, affects the ligase coordination with polβ during substrate-product channeling of repair intermediates at the

downstream steps of BER. In the present study, we extended this work with LIG1/mismatch structures.

In recently published LIG1 structure studies (Tumbale, P. et al. *Nat Commun.*, 2019; Jurkiw, T. et al. *Nucleic Acid Res.*, 2021), below listed six LIG1 (wild-type and mutants) structures have been solved in the absence of metal ion (Mg^{2+}). They were all captured at the step 2 of ligation reaction where an AMP is bound to 5'-P of nick DNA. These LIG1 structures provide evidence that LIG1 is being able to crystallize in step 2 of the ligation reaction in the absence of Mg^{2+} and show that LIG1 does not need Mg^{2+} to effectively crystallize.

- LIG1 wild-type C:G (6P0C)
- LIG1 EE/AA C:G (6P0D)
- LIG1 E592R C:G (6Q1V)
- LIG1 R641L C:G (7L34)
- LIG1 R771W C:G (7L35)

In our present study, we used the LIG1 E346A/E592A (or EE/AA) for crystallization of the protein with nick DNA containing G:T and A:C mismatches. We pre-treated the purified protein with EDTA and obtained the crystals in the absence of Mg^{2+} . We used the same protein and same crystal conditions for all nick DNAs containing cognate A:T and mismatches G:T and A:C. We captured LIG1/A:T and G:T in step 2 and A:C in step 1. Our biochemical characterization of LIG1 EE/AA enzyme for nick sealing efficiency and substrate discrimination against G:T and A:C support the LIG1/mismatch structures. We observed the mutagenic nick sealing of 3'-dG:T and defective end joining of 3'-dA:C mismatch.

In the discussion part of the revised manuscript, we added the information regarding our crystal conditions in the absence of the metal ion and the comparison with previously solved LIG1 structures (see our responses to Points 2 and 5 below).

Point 2:

How do these structures relate to those reported in reference 54? In particular, how might the EE/AA mutation and the absence of Mg^{2+} alter the conformation of the DNA and/or protein and impact the interpretation of the data presented here?

Response: Ref 54 (now is ref 30 in the revised manuscript: Tumbale, P. et al. *Nat Commun.* 2019) demonstrated two-tiered mechanism by which LIG1 fidelity is governed by a high-fidelity (HiFi) interface between LIG1, Mg^{2+} , and the DNA substrate, and 5'-AMP end-processing enzyme Aprataxin suppresses mutagenic and abortive ligation.

In our previous work (Caglayan, M. et al *Nat. Commun.*, 2017), we demonstrated the mutagenic ligation of 8-oxoG:A nick DNA by wild-type LIG1. By Tumbale, P. et al., in addition to the structures listed in our response to Point 1 above, LIG1 was captured at step 2 while engaging with nick DNA containing 8-oxoG:A (6P0E). The authors also resolved the 8-oxoG:A structure using LIG1 EE/AA mutant in the absence of Mg^{2+} .

In order to address the Referee's point here, in the revised manuscript, we prepared Supplementary Figures for the superimposition of all LIG1 structures solved in step 2 in the absence of Mg^{2+} . There was no significant difference in either cognate base pair (A:T and C:G) in the presence or absence of the metal ion under crystal conditions for either wild-type or EE/AA mutant of LIG1 (Supplementary Figure 3a). Furthermore, the superimposition of the LIG1/8-oxoG:A structure with our LIG1/G:T structure showed structural similarity in both structures for the formation of the DNA-AMP intermediate, referring to step 2 of the ligation reaction (Supplementary Figure 12).

Additional questions & comments:

Point 3:

Lines 159-160, Figure 2C: What is the center of rotation for the 3'OH in the A:C structure? Does the ribose adopt a different sugar pucker? What is the distance between the 3'OH and where the 5'P is located in the A:T and G:T structures? Could the different conformation be due to the complex being at step 1 vs 2 of the reaction?

Response:

We prepared a new figure (Supplementary Figure 1 in the revised manuscript) to show the center of rotation for the 3'-OH end in our LIG1/mismatch structures. Our results show that the center of rotation is the C3' atom and that the ribose adopted a different sugar pucker depending on the type of mismatched termini. We observed the C3'-endo sugar pucker in the A:T and G:T structures, while there was no sugar pucker in the LIG1/A:C structure.

Point 4:

Lines 160-165, Figure 3: The positions of nucleotides in the upstream DNA only seem to be shifted in the A:C structure (Fig. 3A), not in the G:T structure (Fig. 3B), but the text indicates that there are shifts in both structures.

Response:

We thank the Referee for bringing this point to our attention. We corrected the text of the revised manuscript as following: Our LIG1/mismatch structures demonstrate that the ligase active site exhibits distinct mismatch-specific conformations and shows significant differences in the positions of the 5'-P and 3'-OH strands at the nick around the upstream and downstream DNA (Figure 3a). In the G:T and A:T structures showing the formation of the 5'-5' phosphoanhydride AMP-DNA intermediate, we observed that 5'-P is closer to the 3'-OH strand of the nick to ensure proper positioning and to seal the phosphodiester backbone (Figure 3b). The distances between the ends of the nick DNA for A:T and G:T were 2.1 Å and 2.8 Å, respectively. In the LIG1/A:C structure, the 3'-OH of the nick DNA was rotated 50° from that of LIG1/A:T. The overlay of both structures demonstrated significant differences in the conformations of the 5'-strand due to clear shifts in the positions of the -1G, -2T, and -3C nucleotides relative to the upstream DNA (Figure 3c).

Point 5:

Lines 186-188. This sentence is confusing. Is the implication that the absence of Mg²⁺ and E346 and E592 in the structures reported here does not alter the structure compared to the G:C structure? What is the RMSD among the structures?

Response: We prepared Supplementary Table 2 in the revised manuscript to compare the RMSD among previously solved LIG1 (wild-type and EE/AA) structures with cognate C:G or damaged 8-oxoG:A (Tumbale, P. et al, 2019) and our LIG1 EE/AA structures presented in this study for cognate A:T, mismatches G:T and A:C. As outlined in our response to Point 2 above, we did not observe any significant difference between all structures solved in step 2 of ligation reaction – C:G versus A:T or mismatched G:T versus damaged 8-oxoG:A – in the absence of Mg²⁺. It is important to note that they have reported LIG1/8-oxoG:A structure only for EE/AA mutant – not wild-type LIG1 – in the absence of Mg²⁺.

R.M.S.D (Å)	LIG1 EE/AA G:T, PDB: 7SXE (present study)	LIG1 EE/AA A:C, PDB: 7SX5 (present study)	LIG1 WT C:G, PDB: 6P0C (ref: 30)	LIG1 EE/AA C:G, PDB: 6P0D (ref: 30)	LIG1 EE/AA 8-oxoG:A, PDB:6P0E (ref: 30)
LIG1 EE/AA A:T, PDB: 7SUM (present study)	0.598	1.046	0.801	0.744	0.812
LIG1 EE/AA G:T, PDB: 7SXE (present study)		0.736	0.766	0.688	0.740
LIG1 EE/AA A:C, PDB: 7SX5 (present study)			1.052	1.005	1.004

In order to further address the referee's point here, we present the below table in our responses. We compared RMSD among our structures with previously solved LIG1 wild-type C:G structure that has been resolved in the crystal conditions supplemented with 200 mM Mg²⁺ using a nick DNA containing 3'-ddC modification. This LIG1/C:G structure also shows step 2 of ligation reaction.

R.M.S.D (Å)	LIG1 WT C:G (Mg ²⁺), PDB: 6P09 (ref: 30)
LIG1 EE/AA A:T, PDB: 7SUM (present study)	0.798
LIG1 EE/AA G:T, PDB: 7SXE (present study)	0.636
LIG1 EE/AA A:C, PDB: 7SX5 (present study)	0.849

Point 6:

Line 217: Is "dGTP:C" a typo? The sentence refers to it as a mismatch. Should it be "dGTP:T" instead?

Response: We thank the Referee for carefully evaluating of our manuscript and figures. We corrected all typos and errors in the revised manuscript.

Point 7:

Line 220: "In consistent with our previous studies..." does not make sense. Should it be "Inconsistent" or "Consistent"?

Response: We thank the Referee for bringing this confusing sentence to our attention. We corrected all such confusing sentences in the manuscript.

Point 8:

Lines 250-251, Figure 7: In the coupled ligation reactions, it appears that pol beta does not insert dA opposite C so that it is not possible to conclude whether or not LIG1 was able to ligate that product (although even when that product is preformed in Figure 6, the ligation reaction is very inefficient, suggesting that both steps are defective).

The labels in Fig. 7B are misaligned. It would also be helpful to have a diagram explaining this experiment.

Response: In Figure 6, we evaluated the substrate discrimination of LIG1 against cognate A:T versus mispairs G:T and A:C in ligation reactions using nick DNA substrates with 3'-preinserted bases that mimic DNA polymerase nucleotide (correct or mismatch) insertion products. Our

results indicate the similar amount of ligation products for both A:T and G:T. However, for the nick DNA containing A:C mispair, ligation efficiency of LIG1 is significantly diminished (~90-fold decrease in the amount of ligation products). These results could mimic the last ligation step after G:T or A:C mismatch incorporation by a DNA polymerase into a gap DNA during DNA synthesis step in almost all DNA repair pathways.

In Figure 7, in order to observe the substrate-product channeling from pol β to LIG1 at the downstream steps of BER pathway, we performed the coupled reactions to measure both enzymes' activities in the same reaction mixture at the same time points. In this assay, the ligation efficiency highly depends on pol β nucleotide insertion fidelity. As we reported previously (Caglayan M., *Nucleic Acids Res.*, 2020), the repair intermediates after pol β mismatch insertions, with an exception of dGTP insertion opposite T (dGTP:T), cannot ligated in the next step by BER DNA ligases. Instead, in our previous work (Kamble P. *et al.*, *J. Biol. Chem*, 2021), we reported that DNA ligase can bind to a gap DNA with a high efficiency and the enzyme attempts to seal 5'-P and 3'-OH ends within a gap repair intermediate, which results in self-ligation of gap DNA itself. In the present study, we further investigated the processing of such repair intermediates and showed that APE1 can remove the mismatched base (dA:C) and can interact with LIG1 at the final BER steps (Supplementary Scheme 3).

To address the referee's point here, we added the following sentence in the result section of the revised manuscript: "...However, we did not observe ligation products in the reaction of pol β dATP mismatch insertion opposite C (Figure 7a, lanes 12-15). This could have been due to inefficient A:C mismatch insertion and reduced base substitution fidelity of pol β , as reported for all possible incorrect base pairings."

We presented diagrams for all repair assays (ligation, coupled, and APE1 exonuclease) we performed in this study as Supplementary Schemes 1 and 2 of the revised manuscript.

Point 9:

The manuscript would benefit from careful proofreading. There are small mistakes throughout.

Response: We thank the Referee for these helpful comments. We substantially revised the manuscript to address the concerns and the Referee's points as summarized below. In order to accommodate the request for editing of the English text, we worked with professional editing service, Springer Nature Author Service, for English language editing. The revised manuscript was significantly edited for grammar, phrasing, and punctuation to improve the clarity, flow and readability of the text.

Reviewers' Comments:

Reviewer #1:

Remarks to the Author:

The resubmitted version of the manuscript from Caglayan's group is significantly improved compare to the original version. It is now well organized, streamlined and clearly written. The critique points are appropriately addressed in the revised manuscript and the point-by-point response.

I currently have one issue:

On page 5, line 114 the authors write: "We observed the C3'-endo sugar pucker in the A:T and G:T structures, while there was no sugar pucker in the LIG1/A:C structure (Supplementary Fig. 1d-f)." The sugar moiety in DNA must have a conformation (pucker). Please, run 3DNA (or other program) to analyze the sugar conformation in the A:C structure. Also, while the sugar pucker in the A:T and G:T structures on Supplementary Fig. 1d-e do look like C3'-endo, please, re-analyze them with a program to be sure.

Reviewer #2:

Remarks to the Author:

The authors have adequately responded to the prior critiques, however the question of the sugar pucker in the Lig1 A:C structure remains somewhat confusing. The authors state that it has no sugar pucker, but this would seem to be an unstable state. Does the electron density clearly distinguish the ribose ring or could it be in a mixture of conformations?

June 21, 2022

Ref: NCOMMS-21-48112A

Structures of LIG1 that engage with mutagenic mismatches inserted by pol β in base excision repair

Point-by-point responses to the Referees' comments are as follows:

Reviewer #1:

The resubmitted version of the manuscript from Caglayan's group is significantly improved compare to the original version. It is now well organized, streamlined and clearly written. The critique points are appropriately addressed in the revised manuscript and the point-by-point response.

I currently have one issue:

On page 5, line 114 the authors write: "We observed the C3'-endo sugar pucker in the A:T and G:T structures, while there was no sugar pucker in the LIG1/A:C structure (Supplementary Fig. 1d-f)." The sugar moiety in DNA must have a conformation (pucker). Please, run 3DNA (or other program) to analyze the sugar conformation in the A:C structure. Also, while the sugar pucker in the A:T and G:T structures on Supplementary Fig. 1d-e do look like C3'-endo, please, re-analyze them with a program to be sure.

Reviewer #2:

The authors have adequately responded to the prior critiques, however the question of the sugar pucker in the Lig1 A:C structure remains somewhat confusing. The authors state that it has no sugar pucker, but this would seem to be an unstable state. Does the electron density clearly distinguish the ribose ring or could it be in a mixture of conformations?

Response to both Reviewers:

We thank both Referees for these very helpful comments regarding Supplementary Figure 1 that shows the comparison of LIG1/mismatch structures showing the position of ribose sugar with Fo - Fc omit electron density maps (3σ) of A:T, G:T and A:C. In order to address the Referees's point, we used 3DNA program and re-analyzed the sugar pucker for all base pairs.

We observed C3'-endo sugar pucker in the A:T, and the C4'-exo sugar pucker in the G:T and A:C structures.

To accommodate this new observation, we revised the text as following "Furthermore, our LIG1/mismatch structures revealed that the ribose adopted a different sugar pucker depending on the identity of the 3'-OH base pair, cognate versus mismatched. We observed the C3'-endo pucker in the LIG1/A:T structure while the LIG1/G:T and LIG1/A:C mismatch structures exhibited the C4'-exo sugar pucker (Supplementary Fig. 1a-c)" Accordingly, we deleted the panels D-F of the Supplementary Figure 1 and added a new reference (Li, S., Olson, W.K. & Lu, X. J. Web 3DNA 2.0 for the analysis, visualization, and modeling of 3D nucleic acid structures. *Nucleic Acids Res*, 47, W26-W34 (2019) to the methods section of the revised manuscript where we describe structure analysis. We agree with the referee's point. The higher resolution is necessary to obtain better electron density map for more accurate position of atom of ribose. Our current resolution of LIG1/mismatch structures are 2.8 Å (A:C) and 3.0 Å (G:T). We would like to point out that this is our first LIG1 structures resolved in our laboratory since we established a new lab in 2019. Our crystal studies are ongoing with improved resolutions.